# A Critical Review of Causal Reasoning Benchmarks for Large Language Models

**Linying Yang[1], Vik Shirvaikar[1], Oscar Clivio[1], Fabian Falck[1]**

[1]Department of Statistics, University of Oxford
Corresponding author: linying.yang@stats.ox.ac.uk

## Abstract

Numerous benchmarks aim to evaluate the capabilities of Large Language Models (LLMs) for causal inference and reasoning. However, many of them can likely be solved through the retrieval of domain knowledge, questioning whether they achieve their purpose. In this review, we present a comprehensive overview of LLM benchmarks for causality. We highlight how recent benchmarks move towards a more thorough definition of causal reasoning by incorporating interventional or counterfactual reasoning. We derive a set of criteria that a useful benchmark or set of benchmarks should aim to satisfy. We hope this work will pave the way towards a general framework for the assessment of causal understanding in LLMs and the design of novel benchmarks.

## Introduction

The recent explosion in Large Language Models (LLMs) has led to widespread consideration of their capabilities across a spectrum of human endeavors (Srivastava et al. 2022). One of these areas is *causal inference*, where multiple benchmarks and tasks have been proposed in an attempt to gauge LLM performance (Zečević et al. 2023; Zhang et al. 2023; Gao et al. 2023). In this paper, we present a critical review of these benchmarks. Building on existing "causal hierarchies" (Pearl and Mackenzie 2018), we taxonomise existing tasks into this hierarchy, highlighting how many of these tasks only address the lowest level of 'causal' reasoning. We then propose a set of criteria that a benchmark should fulfill in order to be useful for evaluating an LLM's causal reasoning capabilities.

**Causal hierarchies.** In this review, our goal is to propose criteria that make a benchmark task suitable for evaluating causal reasoning. We guide this by drawing upon existing taxonomies and hierarchies of causal reasoning in the literature.

Zhang et al. (2023) propose a three-level hierarchy for assessing the causal capabilities of an LLM:

- Type 1: "Identifying causal relationships using domain knowledge."
  Example: Person: I am balancing a glass of water on my head. Suppose I take a quick step to the right. What will happen to the glass?

- Type 2: "Discovering new knowledge from data."
  Example: Here are outcomes for our business if we tried strategy A vs. strategy B: [text about outcomes]. Which one leads to greater success?

- Type 3: "Quantitatively estimating the consequences of actions."
  Example: The patient got doses X and Y of certain medicines last time and reported a 40% decrease in blood pressure. What dose Z should I give the patient this time?

Current LLMs can generally answer Type 1 questions effectively due to their large repository of world knowledge, but struggle to answer Type 2 and Type 3 questions, or can only do so with extensive additional prompting strategies (Zhang et al. 2023).

The *ladder of causation* (Pearl and Mackenzie 2018) divides baseline-level causal reasoning into three rungs:

- Rung 1: "seeing" - describing basic statistical associations, as defined by joint and conditional distributions within the data.
  Example: What is the relationship between this and that?

- Rung 2: "doing" - formalizing the concept of *interventions*, traditionally phrased in terms of do-calculus.
  Example: If I do this, what will happen?

- Rung 3: "imagining" - reasoning about alternative or *counterfactual* scenarios, which may contradict what actually happened.
  Example: If I had done this, what would have happened?

Again, many existing tasks only reach the first rung on the ladder: they measure an LLM's ability to identify associations, but do not introduce the higher-level concepts of Rung 2 and Rung 3 (Zečević et al. 2023). The ladder of causation aligns with the hierarchy from Zhang et al. (2023), and both of them serve as a solid foundation for the discussion of causal reasoning.

## Review of existing work, datasets and tasks

We surveyed the literature and explored repositories and websites that host datasets relevant to causal reasoning in large language models. Starting with review papers such as Zečević et al. (2023) and Srivastava et al. (2022), we collected a comprehensive list of existing datasets and tasks (**39**

in total) used to assess LLM performance in causal reasoning. All benchmarks can be found on this Github repository: https://github.com/linyingyang/CausalReasoningLLM.

## Benchmarks for lower levels of causal reasoning

We find that a significant proportion of existing tasks are benchmarking 'causal parrots', failing to evaluate how well LLMs are capable of causal reasoning and resorting to pre-existing knowledge.

**In-context causal relation identification and extraction.** The first group of datasets used for assessing causal reasoning capabilities consists of human-annotated causal relations. In tasks built on these datasets, a context of facts (such as real life events) or fantasy elements is provided. Along with it, multiple choices of causal relations are presented to the LLM, or the LLM is asked to identify and select the cause or effect event from its provided context.

Datasets along this line of research include MAVEN-ERE, e-CARE, CausalTimeBank(Gao et al. 2023), EventStoryLine (Caselli and Vossen 2017; Gao et al. 2023), Tuebingen cause-effect pairs dataset (Kıcıman et al. 2023),COPA (Gordon, Kozareva, and Roemmele 2012), ChatGPT Causal Reasoning Evaluation, crass ai (Frohberg and Binder 2021), causal judgment (Srivastava et al. 2022),Causal Discovery (Causal Parrots), and Knowledge Base Facts) (Zečević et al. 2023). An example from (Gao et al. 2023) is shown in Figure 1.

We argue here that performing well on such tasks does not sufficiently demonstrate the causal reasoning ability of language models. One reason is the inherent design of the tasks as multiple-choice. The LLMs are given limited options to choose from, sometimes only the two options 'cause' and 'effect'. This lack of 'open-endess' restricts the evaluation of whether LLMs can identify possibilities in a more abstract sense. In other words, given that many of these datasets are directly crafted from basic NLP tasks, by abusing the underlying correlation structure of the data, such as calculating similarities between options and questions in a vector space (e.g., cause-effects pairs in Wang et al. (2022)), LLMs can achieve good performance without actually doing any causal reasoning. If this is the case, the good performance could be attributable to spurious language cues in the datasets, since the underlying mechanism of identifying causal relation is merely from the language used. Examples of simple word replacements with less precise words in prompts (for example 'pets' instead of 'cats' in questions about food to be fed to the animal) led to a drop in performance (Li, Yu, and Ettinger 2022), contributing to this hypothesis.

Input: Minutes after a woman was suspended and escorted from her job at the Kraft Foods plant in Northeast Philadelphia, she returned with a gun and opened fire, killing two women and critically injuring a third co-worker before being taken into custody. **Question:** is there a causal relationship between "suspended" and "injuring" ?
**Answer:** Yes

Figure 1: An example of causal relation identification tasks.

Overall, such tasks constrain the space of causal reasoning that the LLM has to act in, neglecting other types of causal relations (e.g., temporal or spacial, correlative, counterfactual, etc.), allowing targeted training for these highly specific contexts, rather than a general causal reasoning ability which we are interested in. Furthermore, since the LLMs are forced to make a selection, they tend to assume causal relationships between events regardless of whether those relationships actually exist, thus not being able to identify 'causal rather than correlative" relationships which is crucial in causal reasoning tasks.

**Commonsense knowledge: a necessary but not sufficient condition for causal reasoning.** Another commonly seen group of datasets and tasks allow and do not exclude the possibility of knowledge retrieval, for example, com2sense (Singh et al. 2021), winowhy (Zhang, Zhao, and Song 2020), tellmewhy (Lal et al. 2021), Neuropathic-pain-diagnosis (Tu et al. 2019; Tu, Ma, and Zhang 2023), moral permissibility, human organs senses, simple ethical questions, goal step wikihow (Zhang, Lyu, and Callison-Burch 2020), (Srivastava et al. 2022), intuive physics(Zečević et al. 2023), ART (Liu et al. 2023).

The motivation of these tasks is to test if LLMs have acquired commonsense knowledge from everyday experience and can draw sound, intuitive inferences like a human, and hence ground causal reasoning on commensense knowledge (e.g., Figure 2 and 3). Note that in some tasks LLMs are not even provided with specific in-context data supporting the causal reasoning (e.g., in Tu, Ma, and Zhang (2023), no context is provided to ChatGPT before authors submit queries), further indicating that LLMs are only performing knowledge retrieval, not causal reasoning in these tasks. We believe that this is a necessary component of causal reasoning, and that causal reasoning in humans likely relies on applied domain knowledge and accumulated experience. However, such tasks still do not demonstrate that LLMs have acquired the abstraction and imagination skills that constitute the higher levels of our causal hierarchies (Zhang et al. 2023; Pearl and Mackenzie 2018) discussed in the Introduction.

## Contextual reasoning and graph-based tasks move towards more complex conceptions of causality

In light of the problems mentioned above, different approaches have emerged to evaluate LLMs in a way that incorporates higher levels of causal reasoning. Recall that the hierarchy from Zhang et al. (2023) described Type 2 as "Discovering new knowledge from data" and Type 3 as "Quantitatively estimating the consequences of actions". Meanwhile, Pearl and Mackenzie (2018) defined Rung 2 as "doing"—formalizing the concept of *interventions*, traditionally phrased in terms of do-calculus—and Rung 3 as "imagining"—reasoning about alternative or *counterfactual* scenarios, which may contradict what actually happened. In this context, we therefore begin to see that a true causal reasoning task should formalize the idea of interventional and/or counterfactual reasoning, calling for the LLM to apply some level of abstraction or even imagination rather than simple retrieval of domain knowledge.

**Story-based contextual reasoning.** Many tasks in this

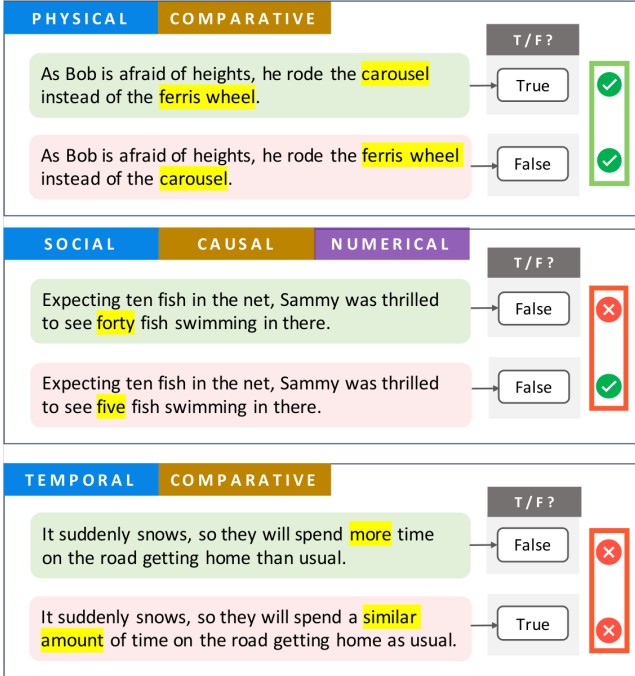

Figure 2: Examples from com2sense (taken from Singh et al. (2021) and Srivastava et al. (2022)).

"A block is placed on one support. What happens if the support is removed?"

Figure 3: An example from intuitive physics (Zečević et al. 2023).

group of datasets are about story-telling, such as Time-Travel(Liu et al. 2023), FinCausal(Mariko et al. 2020), fantasy reasoning(Srivastava et al. 2022), and minute mysteries qa (Srivastava et al. 2022).

These tasks provide a more robust challenge for LLMs by increasing the underlying level of complexity, requiring the language model to evaluate a sequence of several events or statements, discard irrelevant or misleading information, and ultimately synthesize multiple factors to reach a reasonable conclusion. An important innovation across several of these tasks is the use of non-informative or even fictional elements to preclude the application of domain knowledge retrieval. Tasks such as minute mysteries qa (Srivastava et al. 2022) tell a long-form story that with high probability does not exist in the LLM's pretraining dataset, while fantasy reasoning (Srivastava et al. 2022) extend this even further by telling a fantasy story set in a different universe.

One concern with these tasks is the entanglement of 'temporal and spatial order" with causality in the sense that the temporal or spatial order implies the causal relationship. The fact that LLMs are performing well on this task may not be surprising: if event B happens after event A, one may expect LLMs to predict 'A implies B' option, even if the relation is just correlative. This is a common problem across many of these datasets, but is not necessarily alleviated just by in-creasing the length of the description.

The FCR task (Yang et al. 2022) presents a scaling of complexity on multiple stages, where causal relations are categorized as "enable, prevent and cause". Three tasks are embedded within it: 1) a binary classification task to predict whether a given text sequence contains a causal relation; 2) a joint event extraction and fine-grained causality task for identifying text chunks describing the cause and effect, respectively, and which fine-grained causality category it belongs to, and 3) a question-answering task with more challenging "why-questions" and "what-if questions". Though some of these questions can likely still be answered by knowledge retrieval, 2) and 3) move beyond simple causal relation identification, demonstrating the idea that a causally-capable LLM should be able to perform at multiple levels of the causal hierarchy.

**Simplifications in causal discovery.** Another approach to eliciting higher-level causal reasoning builds upon the use of directed acyclic graph (DAG) models. Some tasks even attempt to evaluate the ability of LLMs to return entire causal graphs, as in Causal Discovery (Causal Parrots)(Zečević et al. 2023) and Corr2Cause(Jin et al. 2023), the Neuropathic pain dataset (Tu, Ma, and Zhang 2023), and the Arctic sea ice dataset (Kıcıman et al. 2023).

The main idea behind Corr2Cause and CLadder is causal graph discovery, aiming at inferring the graph structure from a series of conditional independence statements. Corr2Cause technically meets all three ladders in the "ladder of causation", dealing with interventions and counterfactuals as part of the algebraic graph structure. However, its use of letters as the basic 'algebra' is very different to a reasonable real-world interpretation: one may argue that humans would also fail to find causal relationships based purely on conditional independence statements between even just a handful of variables without knowledge about the underlying algorithm.

CLadder rectifies these concerns by shuffling through real-world 'stories' and replacing the node letters with relevant nouns. To the best of our knowledge, CLadder is perhaps the most advanced causal benchmark available currently, as it holistically tests the LLM's ability to synthesize several different components into a complex causal model, and then interprets the effects of interventions or changes within that model. However, it is possible that CLadder's tasks still allow the LLM to use its pre-existing knowledge to return a causal direction, potentially with this causal direction already being included in the training set. While humans may not perform classical causal discovery algorithms with numerical data, they may still perform more complex reasoning than merely remembering the direction, e.g. performing 'intuitive' experiments such as assessing the effect of increasing the altitude on temperature. In addition, merely adding "imagine a self-contained hypothetical world" to the beginning of the prompt as done in CLadder does not imply an LLM will actually follow the instructions. Overall, this suggests that a causal benchmark entirely based on graph discovery of labelled nodes, while a useful and thoughtful development, may not fully encapsulate the necessary causal reasoning capabilities.

Finally, in the Tuebingen cause-effect pairs dataset of Kıcıman et al. (2023), or the causal discovery task of Zečević et al. (2023), edges are evaluated pair by pair independently. We believe that this might be inappropriate as : 1) while humans might break down a full graph discovery problem into problems of smaller graphs, they might still be able to work with more than two variables at the same time, evaluating the direct and indirect causal relations ; 2) as further outlined in Kıcıman et al. (2023), if the DAG contains the edges $A \rightarrow B \rightarrow C$, but not $A \rightarrow C$, it is not clear whether the LLM should answer the individual question "Does A cause C?" as "Yes", as $A$ is indeed a cause of $C$ albeit indirectly, or "No", as the DAG does not contain the direct edge $A \rightarrow C$. Asking for edges of $A, B, C$ jointly would mitigate this issue.

## Many benchmarks suffer from key design issues

**Unsuitable evaluation metrics.** Although some tasks avoid the above issues by prompting the LLM to explain causal relationships (for example, the Causal Explanation Generation(CEG) task (Gao et al. 2023) which ask LLMs to generate explanations for causal relations between events (see Figure 4), the inappropriate design of an evaluation method can hurt the justification of causal reasoning capability of LLMs. In CEG, standard NLP evaluation metrics like *n-grams* or *ROUGE-L* (Wang et al. 2022; Gao et al. 2023), designed for evaluating how similar the generated explanation is with the 'ground truth' explanation provided in the data, are used to measure how 'accurate' the model-generated explanation is. However, to claim a generated explanation accurately reveals causality does not necessarily mean that it has to be syntactically similar with the labeled sentence. Two different sentences using very different words in different order can still convey the same causal concept. Similar tasks and datasets include CausalBank(Li et al. 2021), TimeTravel (Liu et al. 2023). We also notice that some 'ground truth' explanations in these datasets seem to be incorrect as in e-CARE (see for instance Figure 5). How to appropriately generate accurate causal explanations and evaluate them at large scale is an open research question.

Cause: The assailant struck the man in the head.
Effect: The man fell unconscious. Question: why the cause can lead to the effect?
Answer: Hit to head caused brain disruption, leading to unconsciousness.

Figure 4: An example from CEG task.

Cause: Mary sent an emoticon "crying" to her boyfriend on her cell phone.
Effect: Her boyfriend immediately called to comfort her.
Conceptual Explanation: Emoticons are combinations of characters used to represent various emotions.

Figure 5: Examples of incorrect ground-truth explanations in e-CARE.

**Inclusion of datasets or tasks in the training set.** Similar to the commonsense knowledge retrieval issue, another issue with current tasks is that they might include data that is directly in the training set of LLMs. Indeed, experiments (Kıcıman et al. 2023) have shown that GPT-3.5 and GPT-4 memorised the Tübingen cause-effect dataset, or a large portion of it (Mooij et al. 2016). Some papers rely on datasets existing since several years ago, such as the neuropathic pain dataset (Tu et al. 2019; Tu, Ma, and Zhang 2023; Kıcıman et al. 2023), where the chance of inclusion in the training data is more likely.

This further reinforces the idea that in current literature, we cannot exclude the case that LLMs merely memorise knowledge that can be directly or approximately be returned as answers to prompts on causal relationships, underlining that the exclusion of (approximately) retrievable answers is paramount (Valmeekam et al. 2023). Further, reports on newer versions of GPT obtained near-perfection performance on tasks from an earlier version of a paper (section 4.1 of (Zečević et al. 2023)), and one can simply not disentangle whether these improvements are due to improvements to the model (e.g. more parameters, better training procedure etc.) or a memorisation of the relevant tasks. This suggests that LLMs should not be prompted with any pre-existing datasets except if one can guarantee that they have not been used in an LLM's pretraining dataset.

**Poor data quality in the design of datasets.** We have also observed examples of poor quality in some datasets. For example, a task built on e-CARE(Gao et al. 2023) requires a model to choose a correct hypothesis for a given premise from two candidates, so that the chosen hypothesis can form a valid causal conclusion with the premise. However, as shown in Figure 6, there are examples where making the right decision is even difficult for a human, since the options are not well-crafted.

{"She sells rose seeds for a living.", "ask-for": "cause", "hypothesis1": "Maria plants a lot of roses.", "hypothesis2": "The woman's husband thought she was going to be rich because she said she had a great harvest.", "label": 0}

{"He analyzed the composition of the soil.", "ask-for": "effect", "hypothesis1": "He discovered many different elements.", "hypothesis2": "He did it.", "label": 0}

Figure 6: Examples from e-CARE of poor data quality.

Further, to the best of our understanding, the Forecasting Subquestions task of BigBench evaluates the log-probability assigned by human-generated questions that can be related as 'causes" or preliminary subquestions to another given question where higher is better, but does not contain subquestions that are not causes of the question, where lower is better. This prevents the assessment whether the subquestion is merely taking place before the question without any causal relationship or is causally related to it.

**Datasets and tasks that are not for causality.** It is also of concern that some datasets are, in our opinion, not even related to causal reasoning even though they are labeled so. Some datasets are directly crafted from NLP tasks for assessing LLM language understanding. For example, the *BIGbench entailed polarity* (Karttunen 2012; Srivastava et al. 2022) task evaluates an LLM's ability to detect entailed polarities from implicative verbs, as shown in Figure 7. This is by no means a causal reasoning task. In (Yang

et al. 2022), LogiQA, Dream, RACE are also referred to as causal reasoning datasets, but we believe they are targeted at evaluating language understanding as they seem to assess reading comprehension rather than deduction of causal relationships.

Input: Ed remained to be convinced.
Question: Was Ed convinced?
Answer: No

Input: Ed didn't predict that Mary arrived.
Question: Did Mary arrive?
Answer: Yes

Figure 7: Examples in *Bigbench entailed polarity*.

Some other datasets in BigBench, like the speech detection dataset (assessing whether a given figure of speech is a simile, metaphor, pun, etc.),cause and effect, Indic cause and effect (Srivastava et al. 2022) (mainly assessing language translation), are assessing language understanding rather than causal reasoning. We believe that these datasets are inadequate for evaluating causal reasoning and should hence be excluded for this purpose.

## Conclusion and future work

We believe that building a reliable and robust assessment framework for causal reasoning with LLMs is timely and important. In this work, we collected and reviewed existing datasets and tasks, and pointed out issues of certain datasets in light of the three types of causal capabilities (Zhang et al. 2023) and the ladder of causation(Pearl and Mackenzie 2018). We highlight promising recent trends that present a more satisfactory and holistic evaluation of causal reasoning. Ultimately, following the model of the widely popular General Language Understanding Evaluation (GLUE) framework (Wang et al. 2018) for general NLP evaluation, this work is a step towards the creation of a Causal Language Understanding Evalution (CLUE) framework, consisting of a minimal but exhaustive set of tasks that an LLM should be able to complete in order to be considered successful at causal reasoning.

Synthesizing across simpler and more complex tasks, we note key commonalities that we believe a good benchmark or set of benchmarks should possess. First, in the context of the (Zhang et al. 2023) and (Pearl and Mackenzie 2018) hierarchies, we argue that an effective benchmark must **deal with interventions and/or counterfactuals**, being phrased in causal rather than merely correlative language.

In addition, good performance of LLMs on existing tasks may be accounted to their remarkable data-processing and retrieval capabilities using billions of parameters as opposed to their causal reasoning capabilities (Kaddour et al. 2023). Some tasks offer multiple-choice answers to the LLM (Gao et al. 2023; Gordon, Kozareva, and Roemmele 2012; Srivastava et al. 2022), but this can enable the model to simply calculate a measure of similarity between the options and the initial premise. A good benchmark should therefore be **open-ended** rather than multiple choice.

Causal reasoning as needed for Type 2 and Type 3 questions in Zhang et al. (2023) requires the ability to handle complex and possibly intersecting factors, but many existing benchmarks only ask simple one-step questions (Srivastava et al. 2022; Gao et al. 2023). Benchmarks require **multi-factor scalability**. They should be able to introduce one, two, or several additional factors and see how performance changes as complications are added, evaluating the scaling behavior of LLMs. If a machine truly learns algorithms to understand causal relations, adding one more variable should not make any difference, at least within certain bounds of model capacity. We note the causal chains and natural word chain tasks of Zečević et al. (2023) as a positive example to this end.

Lastly, and perhaps most importantly, a pressing concern is that tasks rooted in real-world scenarios or contexts allow for retrieval rather than reasoning. This applies to any dataset for which we cannot exclude that its underlying causal structure has been (approximately) observed in the pretraining dataset of an LLM. Numerous benchmarks prompt an LLM with a story and then ask for its logical consequences (Srivastava et al. 2022; Gao et al. 2023). For example, "if it's sunny in the morning and I forget my umbrella, what will happen if it rains in the afternoon?" (more examples can be found in fantasy reasoning(Srivastava et al. 2022)). The issue here is that the LLM does not need to actually reason at all: it can simply access its training dataset, which contains millions of stories about weather and umbrellas, and approximately retrieve a response (Valmeekam et al. 2022). Of course, humans also access their memory when answering questions like this, but we would be equally capable of answering a similar question with fictional or non-informative context purely by reasoning causally. To accurately gauge human-level causal reasoning, a benchmark task must therefore **not allow for memory retrieval**, outruling the possibility that an answer has been approximately seen in the pretraining data.

We conclude with the following four *criteria*. We argue that not all of these criteria are 'necessary', but they are at least desirable to better demonstrate the causal reasoning capibility of LLMs:

1. **Causal rather than correlative**: The benchmark should be carefully designed in causal language, to reveal causal, not correlative relations dealing with directional interventions and/or counterfactuals.

2. **Open-ended**: The benchmark should allow LLMs to cover as many causal reasoning possibilities as possible, rather than providing a fixed set list of options.

3. **Scalable**: Rather than simple one-step questions, the benchmark should introduce multiple factors, allowing to gradually increase its complexity while following the same causal structure.

4. **Non-retrievable**: The benchmark should be phrased with non-informative or fictional context, such that answers cannot simply be looked up.

We hope that this review can draw the attention of researchers to the pressing need of constructing suitable datasets for causal reasoning and inference with LLMs.

## Acknowledgments

We thank our anonymous reviewers for useful comments which helped us improve the manuscript. L.Y. is supported by the EPSRC Centre for Doctoral Training in Modern Statistics and Statistical Machine Learning (EP/S023151/1) and Novartis. V.S. and O.C. are supported by the EPSRC Centre for Doctoral Training in Modern Statistics and Statistical Machine Learning (EP/S023151/1) and Novo Nordisk. F. F. acknowledges the receipt of a studentship award from the Health Data Research UK-The Alan Turing Institute Wellcome PhD Programme (Grant Ref: 218529/Z/19/Z).

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
