# OpenReview forum: "A Critical Review of Causal Reasoning Benchmarks for Large Language Models"
_AAAI.org/2024/Workshop/LLM-CP — LLM-CP @ AAAI 2024 Oral_

### Official Review · Reviewer_LiRn · 2023-12-03
**Desiderata of benchmark datasets and tasks for studying LLMs' causal behavior**

**Rating:** 2
**Confidence:** 3

**Review:**

This paper examines the desiderata for benchmark datasets and tasks aimed at assessing the ability of large language models (LLMs) to perform causal inference. It effectively summarizes the four desiderata and categorizes existing benchmarks according to these criteria. The paper serves as a valuable resource for the community in selecting or designing appropriate benchmarks to evaluate LLMs in causal inference tasks.

1. The paper is well-written and easy to understand.
2. It has a significant impact on future research at the intersection of causality and LLMs.
3. There is a citation error on page 2.
4. It would have been more compelling to include experimental results to underscore the significance of the presented claims.

---

### Official Review · Reviewer_D2jq · 2023-12-04
**The paper provides a helpful and comprehensive overview and discussion over recent works in causal reasoning capabilities of LLMs. A discussion on how causal relations are actually materialized in natural language could help to improve the paper further.**

**Rating:** 2
**Confidence:** 2

**Review:**

**Summary**
In their paper the authors present a comprehensive overview of works concerned with measuring causal reasoning capabilities of large language models. The paper is structured in two main parts: first, four criteria are proposed that aim to rule out possible biases stemming from previously observed knowledge or allowing the models to infer answers via correlational ('non-causal') cues between questions and answers. Furthermore, the authors argue that scaling scenarios beyond simple bivariate cases is required to truthfully capture real world relations.

In the second part the authors discuss whether the tasks and benchmarks presented in the reviewed papers are actually able to measure the reasoning capabilities of LLMs with regard to (Pearlian) causality. To this end the authors discuss whether all three rungs of the causal ladder are tested and whether the previously proposed criteria are met.



**Strengths**

To the best of my knowledge the presented review is first to provide a comprehensive overview of paper considering causal reasoning capabilities of large language models. The authors are first to propose benchmark criteria and to discuss and evaluate implementation of those for different datasets and tasks.

1) The authors provide a comprehensive overview over most recent papers concerned with causal reasoning abilities in LLMs. To the best of my knowledge relevant works are cited and the authors are first in their attempt to present a unifying set of quality criteria for NLP causal reasoning benchmarks.
2) The authors discuss the strengths and deficiencies of several benchmarks in detail; providing useful information for judging existing and constructing novel benchmarks.



**Weaknesses**

1) The authors to cover a large range of benchmarks with their presented criteria ('open-endedness', 'scalability', ...). However, the paper might be improved by weighting the importance of adherence to the proposed criteria with the goals pursued by the individual papers. All papers highlighted in 'Positive evolutions' employ a kind of dataset that encodes causal graphs within natural language and therefore naturally adhere to the rather rigid formalism of Pearlian causality. This stands in contrast to the rather ambiguous nature of every day natural language. Constructing a dataset from a causal model will inherently restrict the domain of problems to be considered ('what even is a causal variable?', 'at which level are causal relations expressed?'). On the other hand side, measuring causality within datasets comprised of 'every-day' natural language might be better suited to measure real world performance - although deviating from the strict formalism of Pearlian causality.
2) Given the listed criteria in the paper; the paper might be improved by proposing a possible task (or specific query) fulfilling all of the criteria.
3) The authors state that 39 datasets and task from more than 26 papers where considered. As less than 26 papers are cited it is difficult to judge which specific papers where reviewed.
4) The authors present a scenario where the ground truth causal chain "A->B->C" is given. Answers "A->B" and/or "B->C" are considered correct; while "A->C" is not. Following the discussion of the previous points and assuming strict adherence to Pearlian causality this interpretation might be correct. However, when expressed in natural language "Does A cause C" might be considered an equally good answer, whenever B is only considered to mediate the causal effect. (Interpreting 'Does A cause C' as 'Is there any (indirect) effect of A on C'). Equally we would positively answer on the question "Does turning the key of my car start its motor?", while being fully aware that this process requires a chain of additional/mediating factors (cables, battery, fuel level, ...) which are often omitted in every-day language. As recommended in the first point the paper could be improved by adding a discussion on possible ambiguous interpretations of  causal relations in natural language.



**Minor**

1) Among the proposed criteria the 'open-endedness' criterion (with regard to excluding benchmarks providing fixed sets of answers) seems to be the weakest. The authors point out in their discussion that it mainly seems to be established to avoid undesired correlations between questions and answers. However, it might be abandoned in situations where no correlations exist and/or LLMs are only used as 'oracles' within a larger framework (as discussing by the authors for the 'Corr2Cause' [Jin et al. 2023b] or 'Causal Discovery' [Zecevic et al. 2023] datasets). In cases where providing a confined set of answers is a reasonable option, the authors might instead require 'non-correlationality' and full coverage of possible answers (e.g. cause/effect/no connection/not enough information).
2) The font size of the figures is inconsistent and especially small for figs. 2 and 4. Texts (could be made selectable) and use a uniform font size.
3) Towards the end of page two there seems to be a reference missing.



Given the limited number of pages and despite the mentioned weaknesses the authors provide a helpful and comprehensive overview over recent works.

---

### Official Review · Reviewer_SEVR · 2023-12-05
**This review brings up interesting points for community discussion on the design of causal benchmarks.**

**Rating:** 2
**Confidence:** 3

**Review:**

This paper presents 4 criteria that should be satisified by a causal benchmark, and reviews the state of causal benchmarking and evaluation methods against these criteria.

Pros:
- reasoned proposal for key criteria that should be satisfied by causal benchmarks
- likely to lead to interesting and useful discussion at the workshop

Cons:
- The criteria require more refinement to be operationalizable.  For example, it is not clear how to build a benchmark that is both non-retrievable and also consistent with real-world mechanics.
-  It is not clear that there is only one causal task to be benchmarked.  Do distinct causal tasks require different criteria?  E.g., should tests of Corr2Cause's "pure causal reasoning" task be designed with different requirements as compared to tests of actual causality?

---

### Meta-Review · Area_Chair_FFQh · 2023-12-13

**Recommendation:** 2
**Confidence:** 3

**Metareview:**

The paper has interesting ideas. The paper can generate interesting discussions which is the key idea of a workshop.

---

### Decision · Program_Chairs · 2023-12-14

**Decision:**

Accept (Oral)

**Comment:**

Thank you for submitting your work to the LLM-CP workshop @ AAAI 2024. See you in Vancouver!